# MoCA Domain-Specific Pattern of Cognitive Impairment in Stroke Patients Attending Intensive Inpatient Rehabilitation: A Prospective Study

**DOI:** 10.3390/bs14010042

**Published:** 2024-01-09

**Authors:** Benedetta Basagni, Serena Malloggi, Cristina Polito, Leonardo Pellicciari, Silvia Campagnini, Silvia Pancani, Andrea Mannini, Paola Gemignani, Emilia Salvadori, Sara Marignani, Fabio Giovannelli, Maria Pia Viggiano, Bahia Hakiki, Antonello Grippo, Claudio Macchi, Francesca Cecchi

**Affiliations:** 1IRCCS Fondazione Don Carlo Gnocchi, 50143 Firenze, Italy; bbasagni@gmail.com (B.B.); serena.malloggi@unifi.it (S.M.); crpolito@dongnocchi.it (C.P.); leonardo.pellicciari@gmail.com (L.P.); spancani@dongnocchi.it (S.P.); amannini@dongnocchi.it (A.M.); pgemignani@dongnocchi.it (P.G.); emilia.salvadori@gmail.com (E.S.); smarignani@dongnocchi.it (S.M.); bhakiki@dongnocchi.it (B.H.); antonello.grippo@unifi.it (A.G.); claudio.macchi@unifi.it (C.M.); fcecchi@dongnocchi.it (F.C.); 2Department of NEUROFARBA, University of Florence, 50143 Firenze, Italy; fabio.giovannelli@unifi.it (F.G.); mariapia.viggiano@unifi.it (M.P.V.); 3Department of Experimental and Clinical Medicine, University of Florence, 50143 Firenze, Italy

**Keywords:** cognition, cognitive domains, Montreal Cognitive Assessment, post-stroke cognitive impairment, prognosis, rehabilitation, stroke

## Abstract

A domain-specific perspective to cognitive functioning in stroke patients may predict their cognitive recovery over time and target stroke rehabilitation intervention. However, data about domain-specific cognitive impairment after stroke are still scarce. This study prospectively investigated the domain-specific pattern of cognitive impairments, using the classification proposed by the Montreal Cognitive Assessment (MoCA), in a cohort of 49 stroke patients at admission (T0), discharge (T1), and six-month follow-up (T2) from subacute intensive rehabilitation. The predictive value of T0 cognitive domains cognitive impairment at T1 and T2 was also investigated. Patients’ cognitive functioning at T0, T1, and T2 was assessed through the MoCA domains for executive functioning, attention, language, visuospatial, orientation, and memory. Different evolutionary trends of cognitive domain impairments emerged across time-points. Patients’ impairments in all domains decreased from T0 to T1. Attention and executive impairments decreased from T0 to T2 (42.9% and 26.5% to 10.2% and 18.4%, respectively). Conversely, altered visuospatial, language, and orientation increased between T1 and T2 (16.3%, 36.7%, and 40.8%, respectively). Additionally, patients’ global cognitive functioning at T1 was predicted by the language and executive domains in a subacute phase (*p* = 0.031 and *p* = 0.001, respectively), while in the long term, only attention (*p* = 0.043) and executive (*p* = 0.019) domains intervened. Overall, these results confirm the importance of a domain-specific approach to target cognitive recovery across time in stroke patients.

## 1. Introduction

Stroke is one of the main causes of death and functional disability in the adult population [1]. Stroke patients frequently present, together with motor outcomes, persistent cognitive function deficits over time, even in the case of mild strokes [2,3]. It is well known that cognitive deficits negatively impact stroke rehabilitation outcomes and have been associated with lower possibilities of long-term functional recovery, social restoration, and professional reintegration [4,5]. Post-stroke cognitive impairment is present between 46% and 61% of stroke survivors, depending on the assessment tool used, and the odds of having severe cognitive impairment is higher among the stroke survivors compared to non-stroke persons [6]. For these reasons, several studies pointed out the relevance of the early detection of post-stroke cognitive impairments to plan suitable and tailored rehabilitation programs for patients [7,8,9,10]. In parallel, in the literature, the importance of using reliable and valid tools which can help clinicians predict mid- and long-term outcomes in stroke patients is highlighted [7,8].

In this regard, the Montreal Cognitive Assessment (MoCA) [11] is a reliable, valid, and sensitive screening tool for neuro-cognitive deficits detection in cerebrovascular pathologies [12,13,14], even at an early stage, with good predictive capacity for the functional outcome [15,16]. The main strength of using the MoCA on vascular patients is represented by being a rapid and relatively easy-to-use screening tool while providing at the same time a more in-depth assessment of executive functions and the introduction of more demanding visual construction tasks, with respect to the Mini-Mental State Examination (MMSE; [17]) (i.e., another widespread brief cognitive screening tool). The MoCA score is composed of 30 points for items categorised into 6 domains (memory, executive functioning, attention, language, visuospatial, and orientation). Items in each domain yield individual index scores, providing an opportunity to make use of domain-specific test items in characterising cognitive profiles. Two alternative versions of the MoCA (B, C) are available, all of them developed to replace items of the original MoCA version (MoCA A) with similar elements in order to limit possible “learning effects” in longitudinal assessments [18]. The MoCA targets the cognitive domains that can be affected in individuals after stroke: impairments in attention and executive functions appear prevalent after stroke, but also memory, language, and perceptual–motor functioning deficits are reported immediately after the event [2,3].

If the MoCA can be a suitable initial screening, a secondary in depth-analysis of the specific domains affected for each patient should be addressed for the planification of the cognitive rehabilitation treatment [19]. An in-depth neuropsychological assessment is indeed promoting a domain-specific perspective when evaluating the main cognitive outcomes of stroke, to help clinicians in targeting their intervention and maximising the patients’ cognitive recovery [20,21,22]. In fact, studies conducted on global cognitive scores do not consider the different cognitive profiles that can occur after a stroke, such as a prevalent alteration of linguistic skills in patients with left hemispheric lesion or the presence of alterations of visuospatial abilities in patients with right hemispheric lesion, potentially making very different profiles appear similar. In these regards, a recent review by Mole and Demeyere [20] also highlighted that the use of domain-specific cognitive assessment is preferable when researchers are aiming to predict participation and functional activity in stroke patients, with respect to a global cognitive assessment.

A domain-specific approach primarily allows to better characterise patients. Furthermore, it allows to better evaluate the change over time in cognitive functions, which may present different evolutionary patterns to take into account within the rehabilitation plans [22]. Unfortunately, data about the time course of domain-specific cognitive impairment after the stroke are still quite scarce [21,22,23,24]. In fact, available studies mainly focused on the time course of global cognitive impairment in the months after the stroke occurrence [25,26]. The few studies assuming a domain-specific perspective documented an improvement trend in attention, executive functions, and perceptual and verbal fluency abilities from the acute stroke phase to three months later [22,27,28]. However, those studies are retrospective [22], with a limited sample size [22,28], and adopted heterogeneous tools to evaluate the cognitive functioning in stroke patients. 

Within this framework, this work attempts to provide some clinically transferrable information regarding the evolution of cognitive domains of post-stroke patients over time. While preserving a domain-specific perspective, the use of the MoCA test is intended for a first screening potentially more inclusive toward different clinical settings. Additionally, insights on the MoCA cognitive domains mostly influencing a global cognitive recovery on the short- and mid-term post-stroke were provided. The data used were obtained from the RIPS (inpatient rehabilitation post-stroke) study, a multicentre prospective study, including four Italian inpatient rehabilitation Units [29]. RIPS aimed to study multiple features and outcomes in a prospective cohort of consecutively recruited patients addressing intensive inpatient rehabilitation after a stroke, to investigate which features recorded from the multidimensional assessment performed at admission to intensive post-stroke rehabilitation are independent predictors of the functional outcome at discharge. 

In the present study, we propose to prospectively assess, through a domain-specific evaluation using the MoCA classification, the time course of post-stroke cognitive impairments at discharge from post-acute intensive inpatient rehabilitation and after six months from the event (primary objective). Additionally, as secondary objectives, we assess which cognitive domains at admission mainly predict the level of post-stroke cognitive impairment both at discharge from the rehabilitation unit and six months after stroke onset. The final aim is to provide information of clinical relevance for the daily practice, based on a screening tool (MoCA) that could be easily and relatively rapidly applied in diverse clinical settings.

## 2. Materials and Methods

The current study was performed according to the STROBE guidelines [30]. It is a secondary study of a multicentre, observational, and prospective study investigating predictors of stroke outcomes at discharge from inpatient post-stroke rehabilitation (RIPS study) [29]. The RIPS study was carried out in four intensive rehabilitation units of the Fondazione Don Carlo Gnocchi (Firenze, La Spezia, Massa, and Fivizzano). The study protocol was registered on ClinicalTrials.gov (Registration No: NCT03866057) and it was approved by the local ethics committees of each centre (Firenze: 14513; La Spezia: 294/2019; Massa and Fivizzano: 68013/2019). Participants or their legal representatives signed a written informed consent before starting any procedure. The study was conducted following the principles of the Declaration of Helsinki.

A graphical representation of the whole methodology is presented in Figure 1.

### 2.1. Participants

All subjects admitted to either of the four rehabilitation units from December 2019 to December 2020 were systematically assessed for eligibility. Participants meeting the following criteria of the RIPS study were included in the present study: (1) adults (age over 18 years), (2) time from acute event to admission not exceeding 30 days, (3) first-ever admission to the rehabilitation centre for the considered condition, and (4) providing informed consent. Patients were excluded if admitted to the severe acquired brain injuries intensive rehabilitation units, because of a severe haemorrhagic or ischemic stroke, with disorders of consciousness states, and critical clinical care conditions.

In the present work, patients were additionally screened excluding those unable to complete the MoCA or with incomplete MoCA at admission (T0), discharge (T1), or follow-up (T2) were also excluded from the analyses.

### 2.2. Intervention

The rehabilitation intervention performed in all the units involved in RIPS was defined in an ICP based on the American Heart Association/American Stroke Association (AHA/ASA) guidelines [31], and was developed and tested in a previous pilot study [32]. The standardised rehabilitation assessment and the personalised process of care provided, according to the national requirements, at least three hours per day of specific rehabilitation. Physiotherapy, cognitive therapy, speech and dysphagia therapy, occupational therapy, assessment, and training in the use of aids were included. The individual rehabilitation plan was revised based on systematic screening at admission, weekly team revisions, and emerging needs at any time during the rehabilitation stay. As needed, psychological support to the patient and/or family was also provided. 

As to cognitive rehabilitation, all patients whose screening suggested an impairment in one or more cognitive dimensions underwent a comprehensive targeted cognitive assessment and received cognitive rehabilitation. As a general rule, the treatment was delivered by a speech therapist, according to the AHA/ASA guidelines [31]; the standard protocol included one-hour sessions, five times a week, ending only at full recovery or at discharge [29].

### 2.3. Assessment

Participants assessement in the original study addressed different domains, namely: demographics, clinical and nursing complexity, neurological profile, functional evaluation, neuropsychological profile, neurophysiological profile, and genetic analysis. Specifically, the detail of the main variables variable for the identified domains is reported in Appendix A.

In the present study, only cognitive functioning of stroke patients was used for analyses and it was evaluated through the Montreal Cognitive Assessment (MoCA) [10] administered in the RIPS study. The MoCA consists of 12 subtasks exploring different cognitive domains: alternating trail making, cube/rectangle copy, clock drawing, forward and backward digit span, vigilance, serial 7s, sentence repetition, verbal fluency, naming, abstraction, memory, and orientation. The total score is obtained by summing the scores attributed to each subtask and it ranges from 0 (indicating the worst cognitive function) to 30 (indicating the best cognitive function) points. A score < 26 is indicative of cognitive impairment. The Italian version of MoCA (available at https://www.mocatest.org/, accessed on 1 November 2019) was administered in this study. A correction for the performance adjustments concerning the patient’s age and educational level is available in the Italian normative data (Aiello et al. [33] for MoCA version A; Siciliano et al. [18] for MoCA versions B and C). 

For the follow-up, the telephonic version of MoCA (T-MoCA) [34] was adopted for patients who could not be physically present in the hospital during the COVID-19 pandemic. T-MoCA is a modified version of the MoCA administered by phone, which excludes items requiring visual stimuli and pencil-paper drawing. The T-MoCA total score is obtained by summing the scores attributed to each subtask: the global score ranges from 0 (indicating the worst cognitive function) to 22 (indicating the best cognitive function) points, with a score > 19 indicative of cognitive impairment.

Concerning the primary objective, domain-specific MoCA subtests were used [18,33,35], namely:Executive Functioning: this cognitive domain is investigated by three tests: (a) an alternation task adapted from the trail-making B task, (b) phonemic fluency, (c) a verbal abstraction task;Attention: investigated with three tests: (a) serial backward subtraction, (b) letter detection by tapping, (c) forward/backward digit span task;Language: assessed through two tests: (a) naming of three images of low-familiarity animals, (b) repetition of two syntactically complex sentences;Visuospatial: composed of two tests: (a) three-dimension cube copy, (b) clock drawing task;Orientation: composed of a single task in which the patient is asked to answer specific questions over time and place;Memory: consisting of a single memory test composed of delayed recall of five nouns after approximately five minutes from a learning trial.

Regarding the secondary objectives, the above-mentioned MoCA cognitive domains were evaluated with respect to the global level of cognitive functioning (using the MoCA total score as outcome metric) at both discharge and 6-month follow-up, representing short-, T0, and mid-term, T2, outcomes.

### 2.4. Procedure of Data Collection

Participants’ demographic characteristics were retrieved from clinical records. Upon admission to the rehabilitation unit, MoCA (version A) was administered by a neuropsychologist or by a speech therapist to all patients included in the study. At discharge, the MoCA test (version B) was administered again. Six months after the onset, patients were recalled for a follow-up visit and patients who accepted the invitation were administered MoCA (version C). Due to the spread of the COVID-19 pandemic, in some cases, follow-up data have also been collected through telephonic interviews including the remote version of MoCA (T-MoCA) [34].

### 2.5. Statistical Analyses

Statistical analyses were run using SPSS software (version 28.0 for Windows; SPSS Inc., Chicago, IL, USA; 2004). To describe the included sample, i.e., the sample of patients with complete T2 evaluation, descriptive statistics were performed according to the type of variables (numerical or categorical), and, for numerical variables, according to the normality of the data distributions assessed through the Shapiro–Wilk test. Specifically, mean and standard deviation (std), median and interquartile range [IQR], and absolute frequencies with percentage were calculated for normal continuous, non-normal continuous, and categorical variables, respectively. Comparisons of repeated measures of numerical variables over time were performed through the ANOVA test for repeated measures or the Friedman test, in case of normality or non-normality of the distributions, respectively.

At first, to evaluate a potential selection bias on the sample of this work selected from the data collected in the RIPS study, a comparison among clinical characteristics of included and excluded samples was performed. In particular, Chi-squared test (or Fisher test when appropriate), *t*-test, or Mann–Whitney test were performed for comparison of categorical, normally distributed numerical, and non-normally distributed numerical variables, respectively. 

For all subsequent analyses, both dependent and independent variables were dichotomised considering a compromised cognitive function in case of equivalent scores lower or equal than 1, normal cognitive function elsewhere. According to Capitani and Laiacona [36], raw scores may be classified into five ranges corresponding to five categories (0, 1, 2, 3, and 4). An equivalent score of 4 indicates above-median performance (>50 percentile ranks), a score of 0 indicates an impaired performance corresponding to the worst 5% of the normative sample, while the equivalent scores 1, 2, and 3 partition the intermediate ranges (between cut-off and median values). Thus, an equivalent score of 1 indicated a moderately impaired performance. 

The contributions of each MoCA subtest at T0 on the outcomes at discharge (T1) and follow-up (T2) were investigated first through univariate analyses, then with multivariate ones. More specifically, Fisher tests and logistic regressions were performed for univariate and multivariate analyses, respectively. Only those variables significantly associated with the outcome were included in multivariate logistic regressions. Given the relatively small groups, the regressions were performed with backward stepwise selection based on Wald’s coefficient. 

Lastly, on the subgroup exclusively reporting MoCA domains in the three timepoints, multiple time comparisons over time of dichotomous variables were performed through Cochran’s Q test. In the case of statistically significant groups, pairwise comparisons were also performed.

For all statistical analyses, a *p*-value < 0.05 was considered statistically significant.

## 3. Results

### 3.1. Participants 

In the RIPS study, 234 patients with acute stroke were enrolled at admission in the rehabilitation unit. Of these, 167 were excluded because unable to complete the MoCA, and/or lacking data on MoCA in T0, T1, and T2 (Figure 2). Therefore, the final sample of the present study included 67 patients who completed the MoCA 6 months after the stroke and entered the analyses both for the outcome at T1 and T2 (Table 1). 

Due to the COVID-19 pandemic emergency, 17 patients were reluctant to have an in-person follow-up visit. For these patients, a telephonic interview was conducted.

Table 1 displays the detailed demographic, clinical, and functional characteristics of the sample. Comparisons between the included and the excluded participants are provided in the Appendix A. Particularly, the study sample is younger and shows lower stroke severity, higher functional independence at admission, as well as a lower frequency of cognitive impairment compared to the excluded one.

Lastly, detailed information concerning the medications and medical history of patients, concerning comorbidities, is provided in Appendix A. Specifically, according to medications, no patient presented anti-parkinsonian medications, 59.7% took antidepressive medications, and 14.9% took antiepileptic ones. Concerning comorbidities, hypertension and endocrine, vascular, and cardiac pathologies were the most frequent, with moderate to extremely severe severity levels.

### 3.2. Patterns of Cognitive Impairment

Table 2 and Figure 3 display the evolution of the domain-specific cognitive impairment on the MoCA test across the three main phases of the study (T0, T1, T2). 

In detail, the number of patients with impairments in all domains, except memory, is higher in T0 with respect to T1, and only in the case of visuospatial and executive domains the number of altered cases is reducing also from T0 to T2. For the attention domain, the number of altered cases remains stable from T1 and T2; as for the orientation and language domains, the number of patients with impairment increases from T1 to T2. 

Concerning the memory domain, the raw score was evaluated, given the absence of normative adjustments. The results showed a significant improvement between T0 and T1 only. 

Lastly, the language subitems of naming and repetition were also investigated, with the result of a statistically significant trend in time of naming, presenting a significantly reduced number of alterations between T0 and T1 (*p* = 0.003) and a significantly increased of altered cases between T1 and T2 (*p* = 0.003).

### 3.3. Domain-Specific Cognitive Predictors of Patients’ Global Cognitive Functioning at Baseline (T0), Discharge (T1), and Follow-Up (T2)

The univariate analyses revealed a statistically significant association between deficits in the visuospatial, attention, language, and executive MoCA domains at admission and the presence of a global cognitive impairment at discharge (Table 3). Concerning follow-up data, an association between the attention, executive, and orientation MoCA domains at admission and the follow-up presence of global cognitive impairment emerged (Table 4).

From the multivariate analyses, we found that the presence of impairments in executive and language domains at admission predicts an altered global cognitive functioning at discharge (Table 5). The regression model explained 48.7% of the outcome variance (Nagelkerke’s R^2^).

Moreover, we observed that the presence of impairments in attention and executive domain at T0 predicts an altered global cognitive functioning at T2 (Table 6). This second regression model explained 38.4% of the outcome variance (Nagelkerke’s R^2^).

## 4. Discussion

The present study aimed to verify the domain-specific trends of cognitive impairments in stroke patients at discharge from a post-acute intensive rehabilitation path and six-month follow-up, using the MoCA classification. Secondarily, we proposed to investigate which are the cognitive domains that, at admission to the inpatient rehabilitation unit (T0), influence the most the probability of presenting a global cognitive impairment at short- (T1) and mid-term (T2) after stroke onset.

Before entering the discussion of the results, it is worth to give a brief description of the sample used. In this study, the sample of patients was retrieved by the RIPS study, with enrolment conducted between December 2019 and December 2020. Unfortunately, the presence of the COVID-19 pandemic severely influenced the conditions of the study, posing important challenges for what concerns follow-up evaluations. In fact, despite actions initially implemented in the study to counteract potential pitfalls in follow-up assessments, such as realising a detailed informed consent for the patients and offering a follow-up visit together with a full examination and free blood exams, the drop-out rate registered was still high (34.2%). In fact, typical factors such as the willingness of the patient to reach the Rehabilitation Hospital again, the absence of family members available to accompany them to the Rehabilitation Hospital, or the difficulty in organisation of transfers from long-term care facilities have been intensified by the presence of the COVID-19 pandemic. Furthermore, the clinical conditions of some patients have worsened, making in-person visits impossible. In the end, in RIPS, of the patients discouraged by the in-person follow-up visit, only 17 subjects (25.4%) could be available for a telephone interview. The consequences of this situation also led to the presence of a selection bias during the definition of the sample of patients for this work. In fact, given the need for patients with assessments both at discharge (T1) and 6-month follow-up (T2), the analysis sample resulted with patients of younger age, higher NIHSS total score, lower mBI total score, and a reduced number of altered cases on the MoCA scale at baseline (T0). For this reason, it is important to read the following considerations in light of these aspects.

Concerning the primary objective of this study, we found that in this prospective cohort of patients addressing intensive inpatient rehabilitation and receiving evidence-based targeted rehabilitation, global cognitive impairment positively evolves over time. Interestingly, with respect to cognitive domains assessed through the MoCA subtests, the evolution presented different trends across the study phases. Indeed, whilst the percentage of patients with impairment in attentive and executive domains were reduced both at T1 and T2 with respect to T0 (suggesting substantially stable improvement over time after the multidisciplinary rehabilitation program), visuospatial, language, and orientation domains show different trajectories over time.

The percentage of patients with visuospatial deficits decreased from baseline to the short term and slightly increased from short-term to mid-term follow-up, nevertheless maintaining a significant reduction from baseline (Table 2). The performance relative to visuospatial subtests of the MoCA is strongly sensitive to the presence of hemineglect, a disorder frequently occurring after brain damage and affecting 20–50% of patients after a first-ever stroke [37,38]. Therefore, the decreasing trend emerging in the visuospatial domain could be in part related to the initial presence of this disorder, which has been reported to have a rapid and generally regular recovery after the stroke onset [39].

Orientation represents the ability to report time, place, and personal data, and is considered an indicator of level of consciousness, as well as a relevant prognostic indicator of functional outcome in stroke patients [40]. In this domain, a reduction in the percentage of impaired cases is observed from T0 to T1, returning to a similar percentage of impaired cases in T2 (Table 2). A possible explanation of these results could be the loss of the specific training effect of daily reorientation sessions, performed during hospitalisation.

In the present cohort, the language domain shows a contrasting evolution trajectory. We observed a non-significant decrease in the proportion of impaired cases from baseline to the short term and a statistically significant increase from the short-term to mid-term follow-up. An analysis a posteriori of the items of the language domain that most contribute to this pattern revealed that the item most involved, in terms of statistically significant changes, was the naming one. Specifically, a statistically significant improvement was registered between T0 and T1, whilst a statistically significant reduction was observed between T1 and T2. A possible explanation of the T0–T1 change is linked to the effect of rehabilitation and of the speech therapy setting proposed during the hospitalisation, in which the lexical retrieval of words is often practiced starting from images. However, despite statistically significant, the T0–T1 change of naming is not kept in the overall cognitive domain of language, probably due to a combined effect of the two subtasks (naming and repetition). Contrarily, the decrease between T1 and T2 registered on naming is also kept on the overall language domain. A recent meta-analysis reported that naming improvement after stroke heavily relies on age at stroke, with younger people (<55 years) having the best gain in naming performance in the mid-term and long-term follow-up [41]. The present study sample has a relatively old median age, and the age at stroke could potentially influence the progressive impairment in naming performance in the mid-term follow-up. A last possible explanation of this naming impairment could be found in the intrinsic limitations of the MoCA naming subtest itself, which includes only a few items, hence not allowing an exhaustive assessment of naming abilities. Furthermore, the naming task can be compromised not only due to difficulties in accessing the lexicon, but also due to perceptual and visual agnosia impairments.

Overall, our results showed that MoCA domain-specific impairments at baseline were highly prevalent in orientation (42.9%) and attention (42.9%). At 6 months, the MoCA domain-specific recovery was highest in executive (89.8%) and visuospatial (83.7%), while lowest in orientation (59.2%) and language (63.3%). Thus, these findings suggest how particularly orientation and language cognitive domains should be further monitored during and after the hospitalisation. Indeed, especially concerning orientation, very few studies have analysed this cognitive domain in stroke patients. Among them, Pedersen and colleagues [42] showed that impairments in this cognitive domain are mainly related to the stroke severity and the presence of comorbidities, documenting also a recovery across the stroke acute phase. Moreover, Alverzo and colleagues [38] reported that patients’ orientation abilities immediately after stroke strongly influence those assessed four months later.

Additionally, the greatest improvements are observed during the hospitalisation period. Hence, the time between T0 and T1 is characterised by an intensive multidisciplinary rehabilitative path for all the patients included in the study. On the contrary, the time between T1 and T2 is characterised by heterogeneous scenarios but generally characterised by a reduced intensity of care. In summary, our results suggest that patients could be at risk of returning to the level of functioning of the stroke acute phase without an adequate rehabilitation program, especially in the language and orientation domains. Our study highlights the importance of setting adequate and regular in-time rehabilitation programs to prevent their further worsening.

These results are in line with the recent literature. As an example, the work of Oh et al., in 2018 [43], using a domain-specific perspective and a specific tool for the cognitive assessment developed by the same authors (Oh et al., 2013, [44]), significant changes in cognitive function were observed over the 2-year period on a sample of 52 post-stroke patients (recordings made in the first week after the stroke and 3, 6, and 12 months later). The pattern of change exhibited a global increase between 0 and 6 months, and then a gradual decrease. This pattern was followed especially for language, attention, and reasoning/abstraction. However, these results should be read in light of differences in both the assessment tool and potentially the rehabilitation treatment. In fact, the information regarding the possible type and duration of the rehabilitation programmes was missing, posing an important query on whether these patients did or did not undergo neurological rehabilitation. Furthermore, the assessment tool of the different cognitive domains used in this study is not the same. However, the comparison of our results with those of Oh te al. [43] at first assessment and six months after the event showed similar results about the poor evolution of the orientation and the positive evolution of attention domains. Additionally, both studies highlighted a constant positive trend of executive functions domain. Contradictory results were found for memory and language domains (significant changes between 0 and 6 months in the study from Oh et al. [43]) and for the visuo-spatial domain (not significant changes in the study from Oh et al. [43]).

In the present study, as secondary objectives, we also focused on the MoCA domain-specific predictive value of global cognitive functioning on the short (T1) and medium term (T2) after stroke onset. Few studies have focused on the mid- and long-term predictive value of a domain analysis in acute cognitive screening. Between these, in the work of Milosevich et al. (2023), acute domain-specific impairments in memory, language, and praxis significantly predicted the overall severity of cognitive impairment at 6 months [24]. On our sample, we found that the language and the executive domains influence the global cognitive outcome in the short term (T1), while both the executive and attention ones present a predictive role over the presence of cognitive impairment in the mid-term follow-up (T2).

The results obtained by Milosevich et al. [24] on the memory domain are of difficult comparison due to the assessment measures used. In fact, whilst in the OCS a more comprehensive memory assessment is proposed, including both free and multiple choice recall aspects, in the Italian version of the MoCA by Aiello et al. [33], the memory score is limited to the free delayed recall. This aspect made also more challenging a comparison of the pattern of memory domain within time in the analyses of the first objective (Table 2). Similarly, regarding the praxis domain, a direct comparison is difficult; in the OCS, ideomotor praxis is evaluated, whilst in the MoCA, only constructional praxis is included.

Impairments in the language domain are common sequelae of stroke [38,45] and can negatively impact functional recovery in the medium term [46]. Even if we did not include patients who were not able to perform the test, thus possibly excluding the most severe aphasia patients, our results confirmed that language abilities at admission are predictors of the cognitive outcome at discharge from hospitalisation, in line with previous literature [47,48]. Coherently with the sample considered in this study, where all patients underwent intensive rehabilitation according to national guidelines, it is not surprising to encounter the significance of language domain in the short term and not in the medium term [24]. This is particularly true considering, as previously mentioned, the speech therapy setting proposed during the hospitalisation.

Concerning the attention domain, we found a predictive role in global cognitive functioning only at mid-term follow-up. This result could be explained by the relatively short time course of language impairment resolution [38,39,49] with respect to that of the attention domain [50,51]. Indeed, speech impairments after stroke have a tendency to spontaneous recovery, mostly remarkable in the first three months after the onset, and faster for ischaemic than for haemorrhagic aetiology [52]. Thus, it could be the case that language impairment has a very high relative weight on cognitive recovery in the early post-stroke phases. Attentional functions would therefore lose relative importance at this stage, whereas their role in global cognitive recovery would emerge at follow-up, after the language deficit improvement.

Executive functions at T0 predict global scores both at T1 and at T2. Disorders in executive functions are one of the most common cognitive consequences of stroke [51,53] and play a critical role in recovery post-stroke [53]. In particular, they are associated with a higher risk of functional dependence [51], difficulties in returning to work, and poor social participation [54]. Moreover, a poor executive function level reduces the patient’s compliance and treatment adherence [55], as well as the ability to learn novel tasks and benefit from rehabilitation [56]. For these reasons, this cognitive domain is generally of great concern to clinicians and researchers involved in cognitive rehabilitation post-stroke. Overall, our results confirm the importance of these cognitive domains in the reorganisation process after a brain injury by showing that higher executive function scores at onset predict a good overall stroke cognitive outcome.

Despite further improvements that could be performed, both in terms of numerosity and selection of the sample, this study is among the few describing the course of MoCA cognitive impairment over time on post-stroke patients undergoing evidence-based reproducible rehabilitation intervention [29,31]. The predictive value of domain-specific cognitive deficits at admission over the presence of cognitive impairment at discharge and follow-up was also investigated, offering insights on cognitive domains obtained through a brief screening test with relatively rapidly and easy-to-use application. Indeed, the use of the MoCA as the main assessment tool of this work was meant with the purpose to provide initial essential information on the affected cognitive domains and their role in time for the global cognitive recovery.

The MoCA has the advantage of offering a more comprehensive analysis of cognitive domains, including executive functions, while still being a brief screening tool. Its administration time is in the order of 10 min, and prior training of the personnel, multiple clinical figures, including speech therapists, medical doctors, and physiotherapists, can administer it [57]. From a numerical point of view, the analysis for the secondary objectives of association of the MoCA cognitive domains with respect the global cognitive recovery of the MoCA can be delicate. In fact, to mitigate potential dependency effects, in the longitudinal associations, given the fact that the sum of domains is actually retrieving the total score, the dichotomised version of each score was used. In these regards, and especially with the final aim of planning the rehabilitation treatment, a complete cognitive assessment, providing a more in-depth prospective assessment provided over time and including less studied post-stroke cognitive impairment effects such as cognitive–motor interference [58], is needed. However, although such in-depth assessment is desirable and should be achieved on a final basis, it can be time consuming as an initial screening approach during both the acute and subacute phases of the stroke. Hence, in line with what is available in the literature concerning acute phases or long-term durations, where global cognitive functioning measures such as the MoCA and MMSE [59,60] or the OCS [24,61] were used, the MoCA was selected.

In conclusion, in this study, we provided a domain-specific perspective when assessing cognitive functioning in stroke patients, still using a brief screening test with wide applicability in different clinical settings. These findings may have relevant implications to maximise patients’ cognitive recovery across time and, complemented with in-depth specific cognitive assessments, to target their long-term intervention. Our study is among the first analysing cognitive domain of post-stroke patients in a controlled a subacute rehabilitation setting, with patients addressing intensive inpatient rehabilitation and receiving evidence-based targeted rehabilitation. The findings showed different trends of evolution over time of cognitive domains in stroke patients, also addressing the domain-specific predictive value on post-stroke cognitive impairment, thus paving the way to further research on the potential effects of targeted rehabilitation strategies of patients with post-stroke cognitive impairment, both in the post-acute and in the chronic phase.

## Figures and Tables

**Figure 1 behavsci-14-00042-f001:**
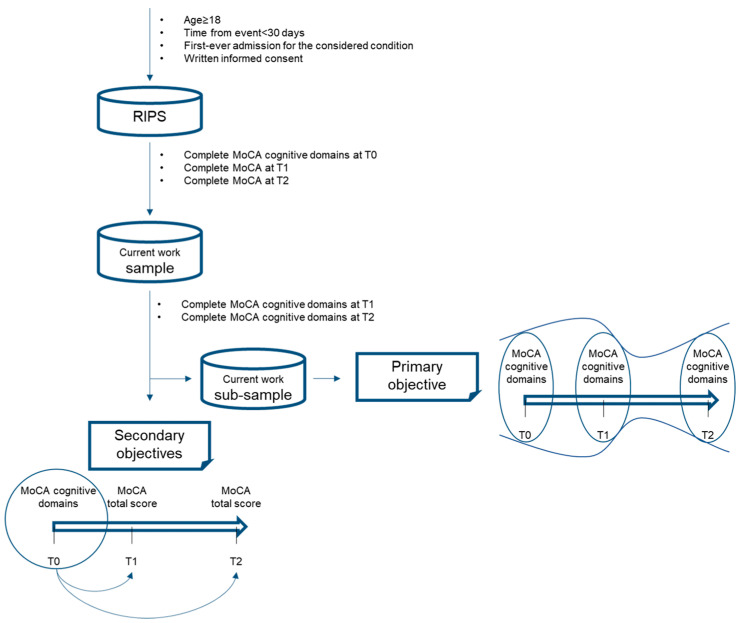
Methodological framework of the study.

**Figure 2 behavsci-14-00042-f002:**
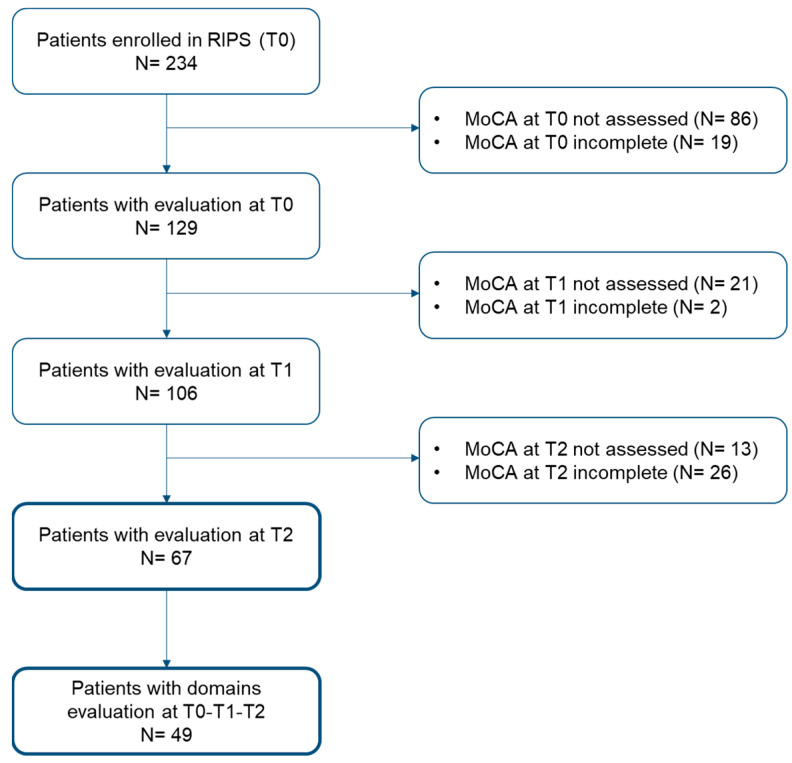
Study flowchart.

**Figure 3 behavsci-14-00042-f003:**
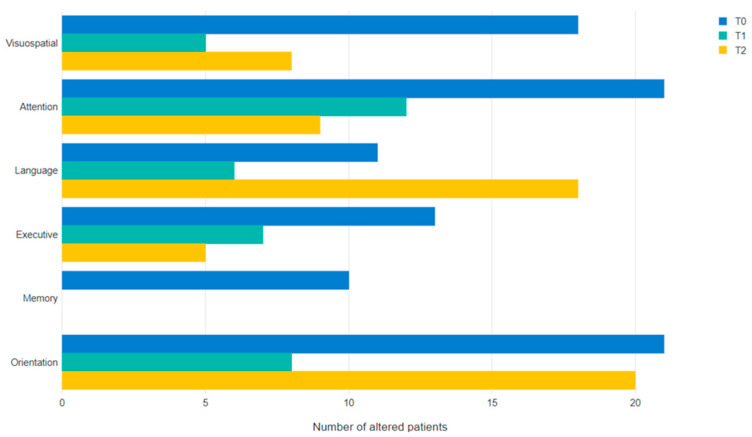
Evolution of the MoCA’s subtests in time.

**Table 1 behavsci-14-00042-t001:** Descriptive analyses of the sample. Statistically significant *p*-values are in bold.

Variables	Median [IQR] or Frequencies At Admission (T0)	Median [IQR] or Frequencies At Discharge (T1)	Median [IQR] or Frequencies At Follow-Up (T2)	*p*-Value
Age (years)	76.0 [16.0]	-	-	-
Gender	Male: 35 (52.2%) Female: 32 (47.8%)	-	-	-
Schooling	8.00 [8.00]	-	-	-
Time from the event (days)	11.0 [9.00]	-	-	-
Type of stroke	Ischemic: 49 (73.1%) Haemorrhagic: 18 (26.9%)	-	-	-
Side of stroke	Right: 35 (52.2%) Left: 24 (35.8%) Bilateral: 6 (9.0%)	-	-	-
Area of the lesion	None: 4 (6.0%) Supratentorial: 52 (77.6%) Subtentorial: 8 (11.9%) Both: 3 (4.5%)	-	-	-
NIHSS score	5.00 [6.00]	2.00 [5.00]	-	**<0.001**
NIHSS item 9 (language)	No aphasia: 53 (79.1%) Mild to moderate aphasia: 9 (13.4%) Severe aphasia: 4 (6.0%) Mute or global aphasia: 0 (0%)	No aphasia: 56 (83.6%) Mild to moderate aphasia: 9 (13.4%) Severe aphasia: 1 (1.5%) Mute or global aphasia: 0 (0%)	No aphasia: 42 (62.7%) Mild to moderate aphasia: 6 (9.0%) Severe aphasia: 1 (1.5%) Mute or global aphasia: 0 (0%)	**0.028**
mBI score	37.0 [45.0]	79.0 [46.0]	93.0 [25.0]	**<0.001**
MoCA_dichotomised	Altered: 34 (50.7%) Normal: 33 (49.3%)	Altered: 12 (17.9%) Normal: 55 (82.1%)	Altered: 25 (37.3%) Normal: 42 (62.7%)	**<0.001**
Length of stay (days)	-	32.0 [20.0]	-	-
Speech therapy treatment	-	No: 25 (37.3%) Yes: 42 (62.7%)	-	-

**Table 2 behavsci-14-00042-t002:** Number of domain-specific altered cases in time. Comparisons between the three time points are reported.

MoCA Subtests	Number of Altered Cases			*p*-Values of Pairwise Comparisons			*p*-Value
	T0	T1	T2	T0-T1	T0-T2	T1-T2	
Visuospatial	18 (36.7%)	5 (10.2%)	8 (16.3%)	**0.001**	**0.012**	1.000	**<0.001**
Attention	21 (42.9%)	12 (24.5%)	9 (18.4%)	**0.034**	**0.002**	1.000	**0.002**
Language	11 (22.4%)	6 (12.2%)	18 (36.7%)	0.513	0.166	**0.003**	**0.004**
Repetition	0: 10 (20.4%) 1: 15 (30.6%) 2: 24 (49.0%)	0: 8 (16.3%) 1: 17 (34.7%) 2: 24 (49.0%)	0: 7 (14.3%) 1: 12 (24.5%) 2: 30 (61.2%)	0.090	0.090	0.107	0.172
Naming	0: 0 (0%) 1: 2 (4.1%) 2: 6 (12.2%) 3: 41 (83.7%)	0: 0 (0%) 1: 0 (0%) 2: 5 (10.2%) 3: 44 (89.8%)	0: 0 (0%) 1: 4 (8.2%) 2: 7 (14.3%) 3: 38 (77.6%)	**0.003**	0.302	**0.003**	**0.031**
Executive	13 (26.5%)	7 (14.3%)	5 (10.2%)	0.125	**0.020**	1.000	**0.018**
Memory (raw score)	10 (20.4%) 1.0 [2.0]	- 2.0 [3.0]	- 1.0 [3.0]	- **0.043**	- 0.480	- 0.189	- **0.045**
Orientation	21 (42.9%)	8 (16.3%)	20 (40.8%)	**0.007**	1.000	**0.014**	**0.003**

Notes. Statistically significant pairwise comparisons are in bold. Comparisons are referred to the patients’ subsample who completed the MoCA in person across the three time points (n = 49). At T1 and T2, no patients with memory domain alteration have been recollected.

**Table 3 behavsci-14-00042-t003:** Univariate analyses for MoCA at T1. Statistically significant *p*-values are in bold.

MoCA Subtests at T0	Outcome: Dichotomised MoCA at T1		
	0: Altered Cognitive Status (N = 12)	1: Normal Cognitive Status (N = 55)	*p*-Value
Visuospatial	Altered: 9 (75.0%) Normal: 3 (25.0%)	Altered: 19 (34.5%) Normal: 36 (65.5%)	**0.021**
Attention	Altered: 10 (83.3%) Normal: 2 (16.7%)	Altered: 23 (41.8%) Normal: 32 (58.2%)	**0.011**
Language	Altered: 7 (58.3%) Normal: 5 (41.7%)	Altered: 8 (14.5%) Normal: 47 (85.5%)	**0.003**
Executive	Altered: 10 (83.3%) Normal: 2 (16.7%)	Altered: 10 (18.2%) Normal: 45 (81.8%)	**<0.001**
Memory	Altered: 2 (16.7%) Normal: 10 (83.3%)	Altered: 10 (18.2%) Normal: 45 (81.8%)	1.000
Orientation	Altered: 9 (75.0%) Normal: 3 (25.0%)	Altered: 26 (47.3%) Normal: 29 (52.7%)	0.114

**Table 4 behavsci-14-00042-t004:** Univariate analyses for MoCA at T2. Statistically significant *p*-values are in bold.

MoCA Subtests at T0	Outcome: Dichotomised MoCA at T2		
	0: Altered Cognitive Status (N = 25)	1: Normal Cognitive Status (N = 42)	*p*-Value
Visuospatial	Altered: 14 (56.0%) Normal: 11 (44.0%)	Altered: 14 (33.3%) Normal: 28 (66.7%)	0.080
Attention	Altered: 19 (76.0%) Normal: 6 (24.0%)	Altered: 14 (33.3%) Normal: 28 (66.7%)	**0.001**
Language	Altered: 9 (36.0%) Normal: 16 (64.0%)	Altered: 6 (14.3%) Normal: 36 (85.7%)	0.067
Executive	Altered: 14 (56.0%) Normal: 11 (44.0%)	Altered: 6 (14.3%) Normal: 36 (85.7%)	**0.001**
Memory	Altered: 5 (20.0%) Normal: 20 (80.0%)	Altered: 7 (16.7%) Normal: 35 (83.3%)	0.751
Orientation	Altered: 19 (76.0%) Normal: 6 (24.0%)	Altered: 16 (38.1%) Normal: 26 (61.9%)	**0.005**

**Table 5 behavsci-14-00042-t005:** Multivariate analysis for MoCA at T1. Statistically significant *p*-values are in bold.

Steps	Independent Variables	B	Standard Error	Wald	*p*-Value	Exp(B)	95% Confidence Interval	
							Lower Limit	Upper Limit
1st step (Nagelkerke’s R^2^ = 0.537)	Visuospatial	−1.370	0.925	2.192	0.139	0.254	0.041	1.558
	Attention	−0.921	1.004	0.842	0.359	0.398	0.056	2.847
	Language	−1.678	0.940	3.188	0.074	0.187	0.030	1.178
	Executive	−2.264	0.963	5.527	**0.019**	0.104	0.016	0.686
	Constant	4.514	1.086	17.264	0.000	91.288	-	-
2nd step (Nagelkerke’s R^2^ = 0.523)	Visuospatial	−1.274	0.900	2.002	0.157	0.280	0.048	1.633
	Language	−1.897	0.893	4.510	**0.034**	.150	0.026	0.864
	Executive	−2.576	0.917	7.892	**0.005**	0.076	0.013	0.459
	Constant	4.146	0.962	18.566	0.000	63.209	-	-
3rd step (Nagelkerke’s R^2^ = 0.487)	Language	−1.818	0.842	4.658	**0.031**	0.162	0.031	0.846
	Executive	−2.936	0.890	10.883	**0.001**	0.053	0.009	0.304
	Constant	3.639	0.832	19.129	0.000	38.035	-	-

Notes. Significant values are in bold.

**Table 6 behavsci-14-00042-t006:** Multivariate analysis for MoCA at T2. Statistically significant *p*-values are in bold.

Steps	Independent Variables	B	Standard Error	Wald	*p*-Value	Exp(B)	95% Confidence Interval	
							Lower Limit	Upper Limit
1st step (Nagelkerke’s R^2^ = 0.384)	Attention	−1.282	0.632	4.113	**0.043**	0.277	0.080	0.958
	Executive	−1.528	0.649	5.543	**0.019**	0.217	0.061	0.774
	Orientation	−1.064	0.637	2.785	0.095	0.345	0.099	1.204
	Constant	2.346	0.607	14.916	0.000	10.439	-	-

Notes. Significant values are in bold.

## Data Availability

The data presented in this study are available on request from the corresponding author.

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
