# Peer review of "MoCA Domain-Specific Pattern of Cognitive Impairment in Stroke Patients Attending Intensive Inpatient Rehabilitation: A Prospective Study"

_behavsci, 2024, doi:10.3390/bs14010042_

Round 1
Reviewer 1 Report
Comments and Suggestions for Authors
The study titled "Domain-specific pattern of cognitive impairment in stroke patients attending intensive inpatient rehabilitation: a prospective study" aims to explore the domain-specific perspective on cognitive functioning in stroke patients and its predictive value for cognitive recovery over time. While the topic is of relevance, the paper suffers from methodological and conceptual shortcomings that need to be addressed for it to meet the standards of rigorous scientific inquiry.
Unclear Definition and Scope of Domains:
The paper lacks clarity in defining the domains under investigation. It is crucial to specify whether the domains referenced are based on the Montreal Cognitive Assessment (MOCA) or other criteria. The use of MOCA should be explicitly stated, and any departure from its domains must be justified. Failure to do so compromises the precision required for scientific communication. I think that If only what is defined in MOCA is used, this should be called the domain of MOCA and not the domain of general cognitive ability. Scientists need this rigor.
Ambiguous Research Methodology:
The methodology section is insufficiently detailed, making it challenging to assess the study's robustness. The absence of information on participant selection criteria, randomization methods, and potential biases raises concerns about the internal validity of the findings. A more comprehensive description of the research design and methodology is essential for readers and reviewers to evaluate the study's reliability.
Statistical Inconsistencies:
The statistical analyses, while conducted using SPSS software, lack clarity in presenting results. The use of the Shapiro-Wilk test for normality assessment is mentioned, but the implications of the results are not discussed. Furthermore, the ANOVA and Friedman tests are referenced without specifying their application to particular data sets. Such omissions undermine the transparency and interpretability of the statistical analyses.
Contradictory Findings and Lack of Synthesis:
The results section presents contradictory trends in cognitive impairments over time. While some domains show improvement, others exhibit deterioration, leading to a lack of a coherent narrative. The paper fails to integrate these divergent findings into a unified interpretation, leaving readers puzzled about the overall impact of intensive rehabilitation on cognitive recovery.
Limited Literature Integration:
The study claims a scarcity of data on domain-specific cognitive impairments post-stroke, yet fails to adequately engage with existing literature that might challenge or support this assertion. A more thorough review of prior research is necessary to contextualize the study's contributions and validate its significance.
Inadequate Discussion of Limitations:
The paper lacks a comprehensive discussion of its limitations, particularly regarding the small sample size and potential confounding factors. Acknowledging and addressing these limitations is crucial for a transparent and responsible scientific discourse.
Overall,
while the paper addresses an important aspect of stroke rehabilitation, it falls short in terms of methodological clarity, statistical rigor, and synthesis of findings. A revision that addresses these concerns is necessary before considering its publication in a reputable scientific journal. There are many existing related studies, so if this paper is not revised strictly, there is a risk that it will be nothing more than a repetition of existing studies.
Comments on the Quality of English LanguageModerate English thought-based expressions need to be modified.
Reviewer 2 Report
Comments and Suggestions for Authors
I have received the research article entitle “Domain-specific pattern of cognitive impairment in stroke patients attending intensive inpatient rehabilitation: a prospective study” by Basagni et al., for evaluation.
In this manuscript, authors have shown the time dependent recovery of global cognitive functioning after stroke in patients. Study is very much interesting and implicated in the society. However, I have few critics for authors to improve the manuscript.
My specific comments are:
1- The number of patients reached to T2 stage are small. I would like to suggest authors to increase the sample number.
2- The background medical history and details about medications are not given in the manuscript.
3- Authors may representation the manuscript in more fascinating way such as incorporating a graphical abstract.
Comments on the Quality of English Language
NA
Author Response
Please see the file attached.

Reviewer 3 Report
Comments and Suggestions for Authors
This research adopted a prospective study design to investigate the domain-specific pattern of cognitive impairments in stroke patients. It revealed that different evolutionary trends of cognitive impairments emerged across time points, and specific items in relation to cognitive impairments had predictive values in functioning recovery after stroke. This study was well-designed and provides exciting and valuable findings to the potential audiences. A few minor changes are suggested.
- The information provided in Table 2 is not directly relevant to the core of this manuscript. It might be more appropriate to put it in the supplementary information.
- The necessity of Figure 2 needs to be clarified. Besides, the label of the x-axis of Figure 2 is missing.
- In line 395, the reference to Milosevich et al. (2023) needs to be included.
- The number of enrolled patients in the final analysis (n=67) is quite limited compared to the initial enrollment (n=234). Please provide some explanation in the discussion.

Author Response
Please see the file attached.

Reviewer 4 Report
Comments and Suggestions for Authors
Hello Dears
Thank you for paying attention to the field of rehabilitation, especially in stroke patients.
In my opinion, the principles of research and essay writing have been well respected.
Author Response
Please see the file attached.

Reviewer 5 Report
Comments and Suggestions for Authors
The main question addressed by the research is the investigation of domain-specific trends in cognitive impairments in stroke patients at discharge from a post-acute intensive rehabilitation path and at six-month follow-up. The study also aims to identify which cognitive domains at admission most influence the probability of presenting a global cognitive impairment at medium (T1) and long term (T2) after stroke onset.
The topic is highly relevant and original in the field, addressing a specific gap by focusing on the domain-specific evolution of cognitive impairments in stroke patients over time.
Compared to other published material, this study adds a detailed analysis of the evolution of different cognitive domains over time in stroke patients. It emphasizes the importance of a domain-specific approach in assessing cognitive recovery, which can inform more targeted rehabilitation strategies.
- Incorporating a control group of patients not undergoing the same intensive rehabilitation could provide a clearer contrast of the effects of the rehabilitation program. Expanding on the types of cognitive assessments used may also provide a more comprehensive understanding of cognitive impairments post-stroke.
- For more information and methods for gathering data, some references are not included and should be added:
. A 2‐year prospective follow‐up study of temporal changes associated with post‐stroke cognitive impairment
. Design, evaluation and prototyping of a new robotic mechanism for ultrasound imaging
. Cognitive-motor interference during functional mobility after stroke: state of the science and implications for future research
. Development of a New Control System for a Rehabilitation Robot Using Electrical Impedance Tomography and Artificial Intelligence
- The authors might consider longitudinal studies with longer follow-up periods to assess the long-term effects of post-stroke cognitive impairments. Incorporating qualitative methods, such as patient interviews, could provide more insight into the patient experience and recovery process.
- The authors could also consider additional visual aids, such as more flowcharts, to depict the study's methodology and patient progress throughout the rehabilitation process.
Author Response
Please see the file attached.

Round 2
Reviewer 2 Report
Comments and Suggestions for Authors
Congratulation